# DiFFPO: Training Diffusion LLMs to Reason Fast and Furious via Reinforcement Learning

## Abstract

We propose **DiFF**PO, **Di**ffusion **F**ast and **F**urious Policy Optimization, a unified framework for training masked diffusion large language models (dLLMs) to reason not only *better (furious)*, but also *faster* via reinforcement learning (RL). We first unify the existing baseline approach such as d1 (Zhao et al., 2025) by proposing to train surrogate policies via off-policy RL, whose likelihood is much more tractable as an approximation to the true dLLM policy. This naturally motivates a more accurate and informative two-stage likelihood approximation combined with importance sampling correction, which leads to generalized RL algorithms with better sample efficiency and superior task performance. Second, we propose a new direction of joint training efficient samplers/controllers of dLLMs policy. Using RL, we incentivize dLLMs' natural multi-token prediction capabilities by letting the model learn to adaptively allocate an inference threshold for each prompt. By jointly training the sampler, we yield better accuracies with lower number of function evaluations (NFEs) compared to training the model only, obtaining the best performance in improving the Pareto frontier of the inference-time compute of dLLMs. We showcase the effectiveness of our pipeline by training open source large diffusion language models over benchmark math and planning tasks.

## 1 Introduction

Reinforcement Learning from Verifiable Rewards (RLVR, Lambert et al. (2024)) has achieved remarkable success in enhancing the reasoning capabilities of Large Language Models (LLMs) (Jaech et al., 2024; Guo et al., 2025). The fine-tuned Large Reasoning Models (LRMs) show drastic improvement in solving tasks which require strong reasoning capabilities, such as Math and Coding (Le et al., 2022; Shao et al., 2024). These LRMs match, and even surpass the performance of the best human players in math contests (Chen et al., 2025). Despite these successes, LRMs are notorious for long inference-time, and overthinking for easy questions (Chen et al., 2024; Su et al., 2025), which limit their applicability to scenarios with low tolerance for latency or inference budgets. An active line of recent research on efficient reasoning (Sui et al., 2025) proposed either training-free early stopping mechanisms, or generation-length-aware RL (Reinforcement Learning) objectives (Kim et al., 2025; Xu et al., 2025). However, the resulting RL fine-tuned models are still fundamentally bottle-necked by the left-to-right autoregressive (AR) decoding in decoder-only transformers (Vaswani et al., 2017), which will inevitably suffer from quadratic inference costs with respect to the length of reasoning traces.

Diffusion LLMs (dLLMs) (Lou et al., 2023; Sahoo et al., 2024; Shi et al., 2024; Nie et al., 2025; Ye et al., 2025), an emerging family of LLMs based on discrete-space diffusion models (Hoogeboom et al., 2021; Austin et al., 2021), have the natural premise of going beyond *left-to-right generations* to *any-order generations* and *multi-token predictions*. Proprietary dLLMs like Mercury (Khanna et al., 2025), Gemini Diffusion and Seed Diffusion (Song et al., 2025) maintain comparable quality to the state-of-the-art AR models while achieving up to 10 times of token throughput. In contrast with the extensive literature on post-training AR models, the relevant study for post-training dLLMs through RL in the literature so far remains rather limited. Whether RL can be used to enhance the reasoning capabilities of dLLMs still remains largely open, and how to design RL algorithms catered for dLLMs to incentivize the capability of base model remains unexplored yet of crucial importance to improve frontier dLLMs.

To narrow this gap, in this work, we design scalable and effective RL algorithms for incentiving reasoning capability of dLLMs. Our contributions in this paper can be summarized as follows:

(1) We first propose an efficient RL post-training paradigm for fine-tuning dLLMs from an off-policy RL perspective. Concretely, we generalize recently proposed algorithms like d1 (Zhao et al., 2025) by proposing to train efficient surrogate policies with more tractable likelihood, as opposed to directly training base dLLMs policies, which requires extensive GPU memory to estimate and is inefficient to compute dLLM's true likelihood. Based on our framework, we propose a new RL algorithm utilizing new surrogate policies by conditioning on additional latents at the response-generation level for better approximation, instead of only conditioning on prompts as in d1. Inspired by classical off-policy RL, we also introduce an importance-sampling correction term to address the distribution mismatch between the surrogate policies and the true dLLMs behavior policies. We provide both theoretical guarantees and strong empirical evidence, achieving evident performance gains over the baseline method, especially for planning tasks (like Countdown, Sudoku) in Figure 1.

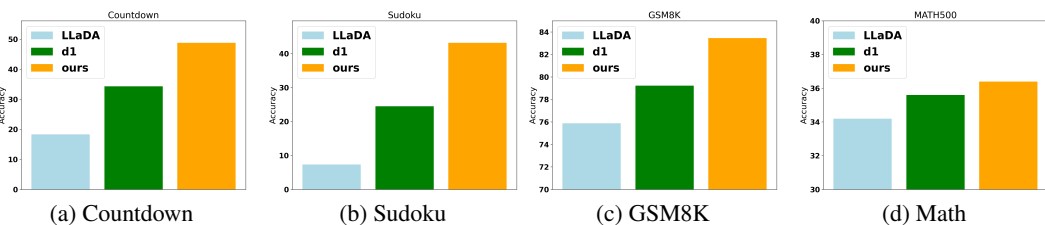

| (a) Countdown | (b) Sudoku | (c) GSM8K | (d) Math |

Figure 1: Benchmark results of RL post-trained models across different math and planning tasks.

(2) Secondly, we innovatively propose to adopt efficient dLLMs samplers in RL post-training. Unlike prior work which post-trains the models based on a fixed (often inefficient) sampler, we directly train the model upon efficiently dLLMs samplers to inventive the base model to reason better and faster. To avoid overfitting to the utilized samplers, we propose to train a prompt-aware inference threshold motivated by the Entropy-Bounded (EB) sampler proposed in Ben-Hamu et al. (2025) via RL as opposed to a fixed threshold across different prompts. This approach naturally leverages the structural property of masked dLLMs for multi-token predictions, and lets the model perform inference based on a predicted "inference threshold" for the prompts. We enable efficient joint RL training of the model and sampler by treating the inference threshold as an additional token, which is thus compatible with our earlier proposed RL framework. We showcase that jointly training the model and sampler can yield improved post-trained models with better performance/accuracy under the same or even less compute than training the model only, unlocking a novel direction for further research to investigate mechanisms between model and samplers in RL post training.

We term our pipeline as **DiFFPO**, **Di**ffusion **F**ast and **F**urious Policy Optimization, for training masked dLLMs to reason not only *better (furious)*, but also *faster* via RL. We demonstrate the effectiveness of DiFFPO in enhancing the inference-time performance of diffusion LLMs on benchmarking math and planning tasks, and clearly push the boundaries of designing scalable RL algorithms towards training efficient and capable LRMs. The rest of the paper is organized as follows: In Section 2, we review the preliminaries of masked dLLMs, samplers, and RL for discrete diffusion models. Section 3 presents our detailed algorithm of DiFFPO, with experimental results provided in Section 4. In Section 5 we discuss other relevant references and conclude with Section 6.

## 2 Preliminaries

In this work, we focus on *masked* discrete diffusion language models for simplicity, which has been shown to achieve the best performance among different formulations of dLLMs. Our framework is general and can be adapted to other form of discrete diffusion models as well. For an introduction to dLLMs, we follow the presentations in MDLM (Sahoo et al., 2024).

**Masked Diffusion Models**. Let [mask] be an extra special token additional to the token vocabulary $\mathcal{V}$, and denote by $\boldsymbol{m}$ the one-hot representation of this mask token. The forward processes of MDLM (Austin et al., 2021; Sahoo et al., 2024) interpolate between the clean data distribution $\boldsymbol{x} \sim p_{\text{data}}(\cdot)$ and a target non-informative distribution $\text{Cat}(\cdot; \boldsymbol{m})$ (the categorical distribution). The

latent variable $z_t$ with $t \in [0, 1]$ in the forward process is:

$$q(z_t \mid x) = \text{Cat}(z_t; \alpha_t x + (1 - \alpha_t) m), \tag{1}$$

where $\alpha_t \in [0, 1]$ is a strictly decreasing function in $t$, with $\alpha_0 \approx 1$ and $\alpha_1 \approx 0$. This process can be viewed as a masking process, because $q(z_1 \mid x) \approx m$. The posterior of the masking process ($t > s$) can be explicitly computed as (see e.g., Sahoo et al. (2024) for a derivation):

$$q(z_s \mid z_t, x) = \begin{cases} \text{Cat}(z_s; z_t) & z_t \neq m, \\ \text{Cat}\left(z_s; \frac{1-\alpha_s}{1-\alpha_t} m + \frac{\alpha_s - \alpha_t}{1-\alpha_t} x\right) & z_t = m. \end{cases} \tag{2}$$

Since $x$ is unknown, MDLMs learn a function approximation $x_\theta(z_t, t)$, and approximate the true posterior with $p_\theta(z_s \mid z_t)$ by replacing $x$ with approximated $x_\theta$ in Equation 2 when $z_t = m$.

**dLLMs efficient samplers**. For masked dLLMs, the backward process needs to choose among unmasked positions which to unmask first. This leaves an extra degree of freedom for designing efficient dLLMs samplers. Most existing works focus on a fixed number of multi-token predictions, such as random sampling or Top-$k$ sampling (Nie et al., 2025) by choosing the position based on some pre-defined scores. The scores are generally chosen from proxy metrics like confidence $s_l = \max_{v \in \mathcal{V}} p_l(v)$ or negative entropy $-\mathcal{H}(p_l)$ of the predictive distribution of dLLMs $p_l \in \Delta^{|\mathcal{V}|-1}$ on the position $l$. These Top-$k$ based samplers are shown to outperform randomly choosing $k$ positions (Kim et al., 2025). Above mechanism is also referred as 'remasking' in (Nie et al., 2025) if we think of the equivalent procedure by unmasking every position and remasking those with low scores.

In this work, our sampling scheme mainly follows the Entropy-Bounded (EB) sampler proposed in Ben-Hamu et al. (2025), which yields better accuracy and efficiency tradeoff than the Top-$k$ sampler that sweeps over different $k$'s. When jointly unmasking a sequence of variables $X = (x^1, \cdots, x^L)$, the EB-sampler computes the entropy of the predictive distribution of dLLMs at each unmasked position as a cost of that position $c(l) = H(p^\theta(x^l \mid X))$, and then unmask all positions in $U$ with the maximum cardinality: $\sum_{l \in U} c(\ell) \leq \gamma$, where $\gamma$ is a predetermined threshold hyperparameter. If $\min_{l \in U} c(\ell) > \gamma$, EB sampler will only unmask the position with the smallest cost. Intuitively, small $\gamma$ will lead to behaviours similar to Top-1 sampling, while large $\gamma$ will allow unmasking of several positions at the same time. Different $\gamma$ forms an inference-time compute frontier for dLLMs.

**RL for dLLMs**. RLVR aims at training models to maximize the expected reward of the policy generations plus an KL-regularization term via Reinforcement Learning (Sutton et al., 1998), i.e.,

$$\mathbb{E}_{c \sim \mathcal{D}, o \sim \pi_\theta(\cdot \mid c)} [r(c, o) - \beta \, \text{KL}(\pi_\theta(\cdot \mid c), \pi_{\theta_{\text{ref}}}(\cdot \mid c))], \tag{3}$$

where $r(c, o)$ is the verifiable reward of generation $o$ with respect to the prompt $c$ (sampled from the population $\mathcal{D}$), such as correctness or unit test success rates, $\text{KL}(p, q)$ denotes the Kullback–Leibler divergence between two distributions $p$ and $q$, and $\beta > 0$ is the KL penalty constant. GRPO (Shao et al., 2024), as a variant of REINFORCE (Sutton et al., 1999) and PPO (Schulman et al., 2017), first samples a group of outputs $\{o_i\}_{i=1}^G$ from the behaviour (old) policy $\pi_{\theta_{\text{old}}}$ under the same prompt $c$, and computes the normalized advantage function $A_i = \left(r(c, o_i) - \frac{1}{G} \sum_{j=1}^G r(c, o_j)\right)/\sigma$, in which $\sigma$ is the standard deviation of the group rewards. Denote $o^{<k}$ as tokens already generated before the $k$th token is decoded within completion $o$, GRPO maximizes the following token level objective:

$$\mathcal{J}_{\text{GRPO}}(\theta) = \mathbb{E}_{c \sim \mathcal{D}, o_i \sim \pi_{\theta_{\text{old}}}(\cdot \mid c)} \sum_{i=1}^G \frac{1}{|o_i|} \sum_{k=1}^{|o_i|} \min\left(\rho_i^k A_i, \text{clip}\left(\rho_i^k, 1 - \varepsilon, 1 + \varepsilon\right) A_i\right), \tag{4}$$

where $\rho_i^k = \pi_\theta(o_i^k \mid c, o_i^{<k})/\pi_{\text{old}}(o_i^k \mid c, o_i^{<k})$ is the token-level likelihood ratio. Notably we omit the KL divergence term in Equation 4 onwards for simplicity, since it has been found to be less impactful in the reasoning performance, see e.g., Magistral (Rastogi et al., 2025). Since $\rho_i^k$ is inefficient to compute for every token in practice, d1 (Zhao et al., 2025) proposed a *mean-field approximation* $\hat{\rho}_i^k = \pi_\theta(o_i^k \mid c)/\pi_{\text{old}}(o_i^k \mid c)$ to replace $\rho_i^k$ in Equation 4, which is easily computable from dLLMs. In essence, the likelihood ratio used in d1 only conditions on the prompts $c$, omitting the already unmasked tokens $o_i^{<k}$ which dLLMs actually condition on during generation.

## 3 DIFFPO

In this section, we present the derivation of our Reinforcement Learning algorithm including the optimization loss function and the update procedures.

## 3.1 Improved techniques in sample efficient model post-training

For the sampler of the dLLM, we first consider a fixed Top-$k$ confidence based remasking, which is how the completions of prompts/questions are generated. We use $k = 1$ for simplicity here to derive the loss objectives without the loss of generality.

**Motivation**. We first revisit the objective of d1 (Zhao et al., 2025) from a likelihood approximation perspective. The loss objective of diffu-GRPO algorithm is:

$$\mathcal{J}_{d1}(\theta) = \mathbb{E}_{c \sim \mathcal{D}, o_i \sim \pi_{\theta_{old}}} \sum_{i=1}^{G} \frac{1}{|o_i|} \sum_{k=1}^{|o_i|} \min \left( \frac{\pi_\theta \left( o_i^k \mid c \right)}{\pi_{old} \left( o_i^k \mid c \right)} A_i, \text{clip}(\frac{\pi_\theta \left( o_i^k \mid c \right)}{\pi_{old} \left( o_i^k \mid c \right)}, 1 - \varepsilon, 1 + \varepsilon) A_i \right),$$
(5)

in which $\pi_\theta \left( o_i^k \mid c \right)$ is the approximation to true conditional token likelihood, and referred by Zhao et al. (2025) as *mean-field approximation*. This approximation has its own advantage of being simple and memory-efficient to compute, making it appealing to be utilized for update at loss optimization, especially when fine-tuning large language models with massive scale.

However, there are two shortcomings in this approximation. **Firstly**, there is a clear *mismatch* between this and the true likelihood, since the dLLMs need to condition on already unmasked tokens, which is omitted in the mean-field approximation. **Secondly**, hardly any theoretical performance guarantees can be made for the policy obtained from minimizing the loss in Equation 5.

To illustrate these limitations, we first change the index from token number to time index to express our loss in a standard way in the discrete diffusion literature, which eliminates the ambiguity in the generative process.[1] For notations, we denote $t_i^k$ to be the actual timestep when the the $k^{th}$ token in completion $o_i$ is unmasked (thus $1 \leq t_i^k \leq |o_i|$), and we use $z_i^t$ to denote the sequence latents at time $t$. For a discrete diffusion model / policy, we can write an equivalent loss to the GRPO objective (Shao et al., 2024) for dLLM as $\mathcal{J}_{dLLM-GRPO}(\theta) =$:

$$\mathbb{E} \sum_{i=1}^{G} \frac{1}{|o_i|} \sum_{k=1}^{|o_i|} \min \left( \frac{\pi_\theta \left( o_i^k \mid c, z_i^{t_i^k - 1} \right)}{\pi_{old} \left( o_i^k \mid c, z_i^{t_i^k - 1} \right)} A_i, \text{clip}(\frac{\pi_\theta \left( o_i^k \mid c, z_i^{t_i^k - 1} \right)}{\pi_{old} \left( o_i^k \mid c, z_i^{t_i^k - 1} \right)}, 1 - \varepsilon, 1 + \varepsilon) A_i \right), \quad (6)$$

which is equivalent to (we now denote $\hat{o}_i^t$ as the difference between $z_i^t$ and $z_i^{t-1}$):

$$\mathbb{E} \sum_{i=1}^{G} \frac{1}{|o_i|} \sum_{t=1}^{|o_i|} \min \left( \frac{\pi_\theta \left( \hat{o}_i^t \mid c, z_i^{t-1} \right)}{\pi_{old} \left( \hat{o}_i^t \mid c, z_i^{t-1} \right)} A_i, \text{clip}(\frac{\pi_\theta \left( \hat{o}_i^t \mid c, z_i^{t-1} \right)}{\pi_{old} \left( \hat{o}_i^t \mid c, z_i^{t-1} \right)}, 1 - \varepsilon, 1 + \varepsilon) A_i \right). \quad (7)$$

The d1 objective can be interpreted as removing all $z_i^{t-1}$ terms in the loss objective Equation 7. However, as shown through generated sample example in Figure 2a and average KL divergence plot in Figure 2b, when the timesteps increase, there is growing mismatch between the true conditional likelihood $\pi_\theta(\hat{o}_i^t \mid c, z_i^{t-1})$ and mean-field approximation $\pi_\theta(\hat{o}_i^t \mid c)$. The average KL divergence between these two distributions grows monotonically with respect to $t$, likely due to the accumulated effect of neglecting the latents.

**Rethinking via off-policy RL**. Given the crude approximations utilized in d1 and the lack of any theoretical guarantee, we propose to formulate the RL task for dLLMs from an off-policy RL perspective by noticing the unique structure of the dLLMs. Since the true likelihood of dLLMs is inefficient to compute, our aim is to bypass it by: (1) find a surrogate policy that is close to the actual dLLM policy, while its conditional likelihood is tractable to compute; (2) optimize the surrogate policy via off-policy RL. A visualization of our pipeline can be found in Figure 3: we assume that the direction of $r$ ($\hat{r}$) encourages the true dLLM (surrogate) policy to generate higher average reward separately. Then if we optimize the surrogate policy following $\hat{r}$, the performances of true dLLM policy under the same model weights $\theta_t$ gets improved if the projections of it to the linear space $\theta_0 + r$, i.e. $\theta_t^\perp$, move along the direction of $r$.

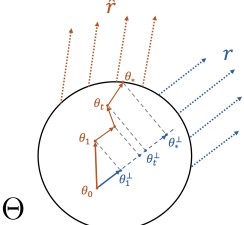

Figure 3: Visualization of off-policy RL for dLLM through optimizing the surrogate policy

---

[1]There is no such ambiguity for AR LLMs, since the generation will be always from left to right.

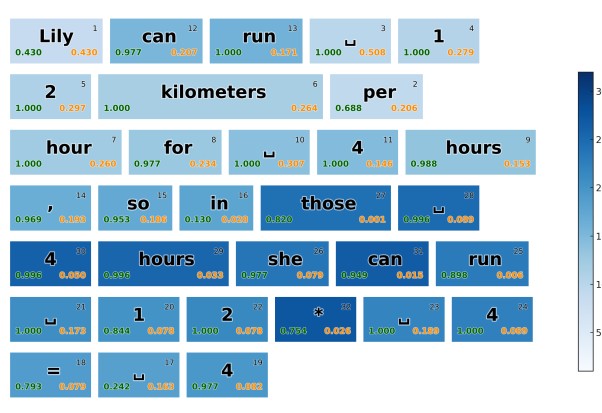

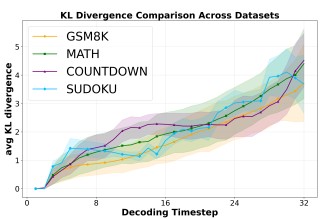

(b) Average KL between the true likelihoods and approximated likelihoods at each timestep.

(a) Sample Generation and Computed Likelihoods: For each block, for example the first one, black part "Lily" represents the unmasked token, green number (0.430) represents the true likelihood when the token gets unmasked, and the orange number (0.430) denotes the mean-field approximation likelihood

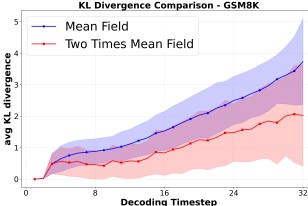

(c) Comparison of 2-MF approximation with baseline mean field approximation.

Before we discuss how we choose the surrogate policies, we highlight that our current pipeline could provide a worst-case performance guarantee on the obtained fine-tuned models:

**Theorem 1.** *Assume two parameterized policies by different model family yet share the same parameters being $\pi_\theta, \hat{\pi}_\theta$ respectively, with $\theta \in \Theta$. Further assuming that reward function is postive and upper bounded, i.e. there exists $M > 0$, such that $0 \le r(c, o) \le M$ for any $c$ and $o$. Then given that $\max_{\theta \in \Theta} \mathrm{KL}(\hat{\pi}_\theta \| \pi_\theta) \le \epsilon^2$, we have that for any policy $\theta \in \Theta$:*

$$\mathbb{E}_{o \sim \pi_\theta(\cdot|c)} r(c, o) \ge \mathbb{E}_{o \sim \hat{\pi}_\theta(\cdot|c)} r(c, o) - \sqrt{2} M \epsilon. \tag{8}$$

The proof is a straightforward application of Pinsker's inequality. Yet, a direct corollary of Theorem 1 implies that, the worst-case of the performance of $\pi^\theta$ can still be guaranteed if $\mathbb{E}_{o \sim \hat{\pi}_\theta(\cdot|c)} r(c, o)$ is large (e.g. after RL optimization) and two policies are close. Back to the dLLM setup, if we take a policy $\hat{\pi}_\theta$ which generates tokens on every position by conditioning only on the prompt, then this is the exact policy `d1` utilized for likelihood approximation, which we have shown to be too coarse with a large KL-divergence. Thus we are motivated to construct *better* surrogate policies to make it closer to the true dLLM policy.

**Additional conditioning latents**. We propose to condition on one additional latent at a randomly sampled timestep $\tau \in [0, T)$ at each optimization step, and consider a new surrogate policy which samples tokens according to the `d1` likelihoods until $t = \tau$. When $t > \tau$, we sample tokens by conditioning on both prompts and latents $z^\tau$. Conditioning on the randomly drawn timestep $\tau$, we define the likelihood of the completion as $\pi_\theta(\hat{o}_i^t \mid c, z_i^{s_\tau(t)})$ where we define a step function $s_\tau$ such that $s(t) = \tau$ if $t > \tau$, and 0 otherwise. We can interpret this as an alternative approximation to the true dLLM likelihood. However, instead of only conditioning on prompts, when $t > \tau$ the generation will also condition on a randomly drawn latent $z^\tau$. We refer to this as Two Times Mean-Field Approximation (abbr. as 2-MF). Compared to `d1` (Zhao et al., 2025), two times mean-field yields a strictly better likelihood approximation. We theoretically characterize its benefits by the following assumptions and theorems.

**Assumption 2.** *Assuming that for any trajectory $z_i^t (t = 0, \cdots, T)$ and three timesteps $s < \ell < t$,*

$$f(s; t) := \mathrm{KL}(\pi_\theta(\hat{o}_i^t \mid c, z_i^{t-1}) \| \pi_\theta(\hat{o}_i^t \mid c, z_i^s)) \ge \mathrm{KL}(\pi_\theta(\hat{o}_i^t \mid c, z_i^{t-1}) \| \pi_\theta(\hat{o}_i^t \mid c, z_i^\ell)) = f(\ell; t), \tag{9}$$

*i.e. $f(\tau; t)$ is a monotonously decreasing function with respect to $\tau \in [0, t]$.*

Assumption 2 assumes that a more recent latent on the same generation trajectory will be more informative than older ones. It is reasonable since the older latents are closer to the non-informative priors. Based on this assumption, we characterize the theoretical benefits of our approximation:

**Theorem 3.** *Under Assumption 2, we have that two times mean-field approximation provides better approximation than only conditioning on prompts, i.e. for any $t$, we have:*

$$\mathbb{E}_{\tau,t} \, \mathrm{KL}(\pi_\theta(\hat{o}_i^t \mid c, z_i^{t-1}) || \pi_\theta(\hat{o}_i^t \mid c, z_i^{s_\tau(t)})) \leq \mathbb{E}_t \, \mathrm{KL}(\pi_\theta(\hat{o}_i^t \mid c, z_i^{t-1}) || \pi_\theta(\hat{o}_i^t \mid c)). \quad (10)$$

*Proof.* The inequality could be directly obtained by tower's rule of expectation and proving that

$$\mathbb{E}_t \, \mathrm{KL}(\pi_\theta(\hat{o}_i^t \mid c, z_i^{t-1}) || \pi_\theta(\hat{o}_i^t \mid c, z_i^{s_\tau(t)})) \leq \mathbb{E}_t \, \mathrm{KL}(\pi_\theta(\hat{o}_i^t \mid c, z_i^{t-1}) || \pi_\theta(\hat{o}_i^t \mid c))$$

holds for any fixed $\tau$. This can be easily obtained from Assumption 2. $\qquad\square$

In Figure 2c, we show the clear empirical evidence on GSM8K dataset that two times mean field approximation indeed yields better approximation to the true likelihood in terms of reducing the KL divergence error. We provide similar results for other datasets in Appendix A.1.

**Off-policy RL via Importance Sampling**. With a better surrogate policy, we now seek to optimize such policy using off-policy RL. Note that the generations are still sampled from the base policy. We can thus utilize these samples to optimize $\hat{\pi}^\theta$ using importance sampling, as for any function $f$, we have:

$$\mathbb{E}_{o \sim \hat{\pi}_\theta(\cdot|c)} f(c, o) = \mathbb{E}_{o \sim \pi_\theta(\cdot|c)} \frac{\hat{\pi}_\theta(\cdot \mid c)}{\pi_\theta(\cdot \mid c)} f(c, o). \quad (11)$$

Then replacing $f$ with the GRPO loss, our DiFFPO loss objective for training the model only is thus:

$$\mathcal{J}_{\text{DiFFPO-model}} =$$

$$\mathbb{E}_{o_i \sim \pi_{\theta_{\text{old}}}, \tau \sim \mathcal{U}[0,T]} \sum_{i=1}^{G} \frac{1}{|o_i|} \sum_{t=1}^{|o_i|} \min(C, \underbrace{\frac{\pi_{\text{old}}(\hat{o}_i^t \mid c, z_i^{s_\tau(t)})}{\pi_{\text{old}}(\hat{o}_i^t \mid c, z_i^{t-1})}}_{\text{IS term}}) \min\left(\bar{\rho}_i^{\,t} A_i, \mathrm{clip}\left(\bar{\rho}_i^{\,t}, 1 - \varepsilon, 1 + \varepsilon\right) A_i\right),$$

$$(12)$$

where $\bar{\rho}_i^t = \frac{\pi_\theta\left(\hat{o}_i^t | c, z_i^{s_\tau(t)}\right)}{\pi_{\text{old}}\left(\hat{o}_i^t | c, z_i^{s_\tau(t)}\right)}$ is the likelihood ratio of the surrogate policy and we also impose a maximum threshold $C$ over the importance sampling ratio to ensure the numerical stablibity.

### 3.2 UNLOCKING BETTER ADAPTIVE MULTI-TOKEN PREDICTION VIA SAMPLER POST-TRAINING

Common practice of RL post training considers a fixed generative model sampler in optimization. In this work, we however investigate on joint training the sampler together with the model weights for achieving the best performance in enhancing the inference time compute frontier.

**Prompt adaptive threshold**. Unlike the EB-sampler (Ben-Hamu et al., 2025) that uses a fixed threshold to decide which token(s) to unmask for all prompts, we propose to use a parameterized function to encode prompt features and output a predicted "inference threshold". We obtain the dLLMs embeddings of the prompts by averaging over the hidden dimension features $f_\theta^l(c)$ of each position $l$, and train a linear mapping $\boldsymbol{w}$ on top of the embeddings (we omit the bias term here for clarity). We also introduce an upper threshold $\gamma_{\max}$, and map each feature to the inference threshold by

$$\gamma_{\boldsymbol{w}}(c) = \gamma_{\max} \, \mathrm{sigmoid}\left(\boldsymbol{w}^\top \left(\tfrac{1}{L} \sum_{l=1}^{L} f_\theta^l(c)\right) + \beta\right), \quad \text{with } \beta = \log \frac{\gamma}{\gamma_{\max} - \gamma}, \quad (13)$$

where $\gamma \in (0, \gamma_{\max})$ is the initial global threshold, since $\gamma_{\boldsymbol{w}_0}(c) = \gamma$ when the initial linear layer weights $\boldsymbol{w}_0$ are set to be all 0. For RL training, we fix a noise level $\sigma$ for Gaussian exploration $\epsilon \sim \mathcal{N}(0, I)$ before applying sigmoid activation and use the perturbed threshold for inference:

$$\gamma(c) = \gamma_{\max} \, \mathrm{sigmoid}\left(\boldsymbol{w}^\top \left(\tfrac{1}{L} \sum_{l=1}^{L} f_\theta^l(c)\right) + \beta + \sigma\epsilon\right). \quad (14)$$

**Joint Training of the model and sampler**. We use $\pi_\theta^w$ to represent the sampler threshold policy which is dependent on both model weights $\theta$ and header weights $w$. To joint optimize the model

and sampler, we apply a trick by treating the predicted threshold as an additional token to unmask at time step 0. Thus, given the group completion $(\gamma_i, o_i)$, we have yield our final DiFFPO loss as:

$$\mathcal{J}_{\text{DiffPO}} = \mathcal{J}_{\text{DiffPO-model}} + \mathbb{E}_{c \sim \mathcal{D}} \sum_{i=1}^{G} \frac{1}{|o_i|} \min\left(\frac{\pi_{\theta_-}^w(\gamma_i \mid c)}{\pi_{\theta_-}^{w_{\text{old}}}(\gamma_i \mid c)} A_i, \text{clip}(\frac{\pi_{\theta_-}^w(\gamma_i \mid c)}{\pi_{\theta_-}^{w_{\text{old}}}(\gamma_i \mid c)}, 1-\varepsilon, 1+\varepsilon) A_i\right). \tag{15}$$

We found that the model learns to generate shorter sequences even without the length penalty involved as in the experiments section, which prevents sensitive hyperparameter tunings. In addition, fixing model weights (i.e. `stopgrad` on $\theta_-$) for the sampler loss in Equation 15 helps both stable training and better performance.

## 4 EXPERIMENTS

To evaluate the performance of the our proposed RL algorithms, we conducted extensive experiments on training base dLLMs with various reasoning tasks and compared their reasoning capabilities in terms of both accuracy and efficiency after RL post-training.

**Experimental Setup.** We choose LLaDA-8B-Instruct (Nie et al., 2025)[2] as our base model since this is the first diffusion large language model whose performance is on par with AR LLMs (Llama3-8B). Improving over such a capable base model is of great interest to the whole dLLMs community. We directly start from the instruct model as opposed to the base model which has not been fine-tuned for instruction following. This avoids any potential confounding effect of us performing additional supervised fine-tuning (SFT) stage which can affect the performance and fair comparison of RL algorithms.

We evaluate all the methods on math and planning benchmark tasks including GSM8K, Math, Sudoku, and Countdown. We use LoRA (Hu et al., 2022) with rank $r = 128$ for parameter-efficient training and also adopt the 4-bit quantized version of the model for efficiency, same as `d1`. Our training is conducted on 8 NVIDIA-A100 40GB GPUs, batch size of 3 per GPU and use gradient accumulation steps of 4. We grid search the best learning rate and clip ratio for each method, and adopt the same all other hyparameters utilized in `d1` (Zhao et al., 2025) like AdamW optimizer for a fair comparison. In addition, we use the same five random reeds for the baseline `d1` and our proposed algorithm, and report the best performing model among different seeds and checkpoints.

For the dLLM inference setup, our experiments are all based on block size 32 and maximum sequence length of 256. We adopt a Top-$k$ confidence based remasking mechanism for inference when training the model only without training the sampler. We choose $k = 2$, the same setup as `d1`, resulting the effective diffusion steps being 128.

**Improved sample efficiency and task performance**. We first showcase the effectiveness of our algorithm when training a DiFFPO model with Equation 12 on math and planning benchmarks.

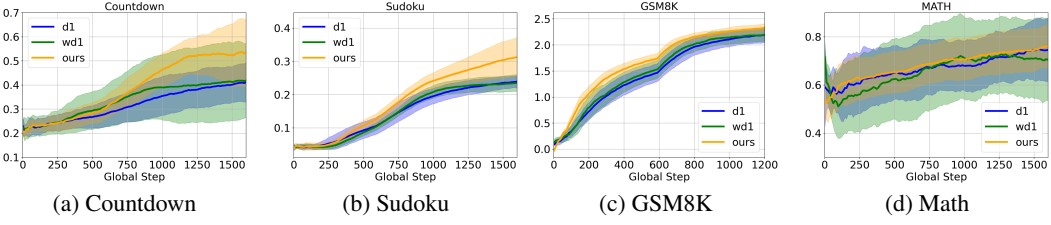

|  (a) Countdown | (b) Sudoku | (c) GSM8K | (d) Math |

Figure 4: Benchmark results (—: `d1`; —: `wd1`; —: `ours`).

We plot the reward as the training progresses in Figure 4. Our algorithm shows a clear margin over `d1` in all tasks, especially in planning tasks like Countdown and Sudoku. Our algorithm also

---

[2]https://huggingface.co/GSAI-ML/LLaDA-8B-Instruct.

outperforms concurrent work like `wd1` (Tang et al., 2025) that also targets at improving d1 on the same reasoning datasets. [3]

| Model | Countdown | | | Sudoku | | | GSM8K | | | MATH500 | | |
|---|---|---|---|---|---|---|---|---|---|---|---|---|
| | Acc. ↑ | Δ | ETs | Acc. ↑ | Δ | ETs | Acc. ↑ | Δ | ETs | Acc. ↑ | Δ | ETs |
| **LLaDA-8B-Instruct** | 18.36 | – | 214.18 | 7.37 | – | 220.45 | 75.89 | – | 211.63 | 34.20 | – | 234.66 |
| + d1 | 34.38 | +16.02 | 176.04 | 24.51 | +17.14 | 233.42 | 79.23 | +3.34 | 143.12 | 35.60 | +1.40 | 228.31 |
| + Two Times Mean-Field | 44.53 | +26.17 | 78.94 | 29.74 | +22.37 | 235.39 | 81.05 | +5.16 | 127.33 | 35.80 | +1.60 | 232.36 |
| + Importance Sampling | **48.83** | **+30.47** | 74.98 | **43.21** | **+35.84** | 244.22 | **83.47** | **+7.58** | 128.95 | **36.40** | **+2.20** | 232.98 |

Table 1: Benchmark results with statistics **(Accuracy, Δ, Effective Tokens (ETs))** under Top $k$ sampler. Δ indicates the accuracy improvement over **LLaDA-8B-Instruct**; positives are highlighted in green.

| Model \Acc. ↑ \Seq Len | Sudoku | | | Countdown | | | GSM8K | | | MATH500 | | |
|---|---|---|---|---|---|---|---|---|---|---|---|---|
| | 128 | 256 | 512 | 128 | 256 | 512 | 128 | 256 | 512 | 128 | 256 | 512 |
| **LLaDA-8B-Instruct** | 10.01 | 7.37 | 6.40 | 17.97 | 18.36 | 21.09 | 68.16 | 75.89 | 79.83 | 26.20 | 34.20 | 35.20 |
| d1 | 23.88 | 24.51 | 25.98 | 27.34 | 34.38 | 37.89 | 72.86 | 79.23 | 78.92 | 33.20 | 35.60 | 37.20 |
| 2MF+IS | **36.43** | **43.21** | **45.70** | **58.98** | **48.83** | **56.64** | **75.74** | **83.47** | **82.26** | **34.80** | **36.40** | **38.60** |

Table 2: Ablations on the maximum sequence length of the model.

We report concrete benchmark statistics in Table 1. The row of '+ Importance Sampling' corresponds to adopting two times mean-field and importance sampling at the same time, while the row of '+ Two Times Mean-Field' corresponds to adopting two times mean-field only. Evidently, our proposed DiFFPO, combining both two times mean-field approximation and importance sampling, yields the most significant performance gain, which both components contribute to.

We also compare the performance difference for the same model checkpoint utilizing different maximum length as in Table 2 (in which we denote '2MF' as an abbreviation for Two Times Mean-Field Approximation, and 'IS' for importance sampling). We found that our DiFFPO trained models stably perform the best.

**Training DiFFPO with efficient samplers**. We also investigate the performance of RL-training on existing efficient samplers to gain the best performance. We tested upon EB samplers (Ben-Hamu et al., 2025), which yields a greater trade-off between accuracy and NFEs compared to the top-$k$ sampler that we used above. We found that the inference threshold $\gamma = 0.1$ yields the best results in achieving the highest reward, on par with smaller $\gamma$'s and top-$k$ (Table 1, $k = 2$), and better than larger $\gamma$'s which will typically yield a smaller average reward even after RL training.

| Model | Sudoku | | Countdown | | GSM8K | |
|---|---|---|---|---|---|---|
| | Accuracy ↑ | NFEs ↓ | Accuracy ↑ | NFEs ↓ | Accuracy ↑ | NFEs ↓ |
| **LLaDA-8B-Instruct** | 10.45 | 201.23 | 22.66 | 212.33 | 80.74 | 98.14 |
| + d1 | 25.88 | 105.39 | 35.16 | 197.05 | 82.34 | 65.71 |
| + **DiFFPO** w/o sampler joint training | 32.23 | 141.14 | 47.27 | 38.80 | 83.47 | 57.68 |
| + **DiFFPO** w/ sampler joint training | **37.16** | **56.80** | **51.17** | **30.45** | **84.00** | **44.00** |

Table 3: Benchmarking planning and math tasks results on both accuracies and NFEs under EB.

**Joint training of the model and the sampler**. Finally we consider joint training of the model and sampler by adding up the model and the sampler losses (Equation 15), to improve the inference-time frontier. In Table 3, we report the optimal checkpoint performance alongside the baseline algorithms. We observe that joint training not only improves the (optimal) correctness, but also uses even less NFEs compared to training the model only.

For a more detailed comparison, we draw the inference-time compute frontier for Countdown for two settings: 1) post-trained DiFFPO model with a fixed top-$k$ sampler (Equation 12); and 2) the

---

[3]Note that since our paper's main focus is on the probability ratio and wd1 focuses on advantage function, it is possible to combine these two methods. We leave the investigation as future work.

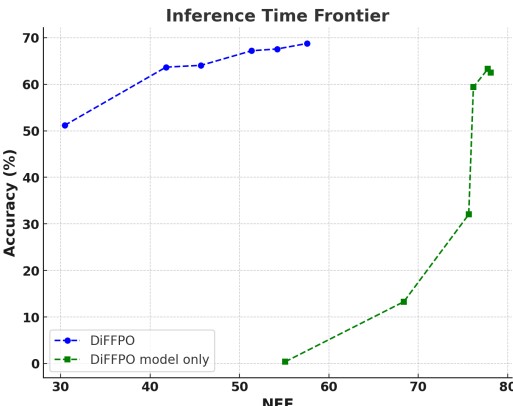

Figure 5: Inference-time frontier obtained by models by DiFFPO with or without sampler training and EB sampler with different inference threshold.

model and sampler jointly trained by DiFFPO (Equation 15), under the same 256 maximum generation length in Figure 5. The dots are obtained by utilizing different base inference threshold (here we choose [0.05, 0.1,0.2, 0.5,1,5]) in EB sampler on two models. The clear advantage of DiFFPO trained models (i.e. with sampler joint training) showcases the importance of choosing proper performant samplers when training the models via RL which is usually overloooked, and the effectiveness of our joint training mechanism in enhancing the inference-time compute frontier.

## 5 RELATED WORKS

We discuss other related works on dLLMs, inference, and RL.

**Discrete Diffusion Models**. In our paper, we mainly focus on masked diffusion models with a discrete-time Markov chain setup (Austin et al., 2021; Sahoo et al., 2024). Other works have pursued different formulations, e.g., through continuous-time Markov chains (Campbell et al., 2022) or flows (Gat et al., 2024; Shaul et al., 2024). Despite our focus on masked diffusion models, our methodology developed in this paper is general and could benefit RL training for these other formulations. Several works have developed unified viewpoints for diffusion models in both discrete and continuous spaces, including Ren et al. (2024); Holderrieth et al. (2024); Sahoo et al. (2025a). The dLLMs family has also been expanded to the multimodal domain, e.g., MMaDA (Yang et al., 2025) and Fudoki (Wang et al., 2025b).

**dLLMs inference**. Most existing works on dLLMs inference have focused on developing more performant samplers, using ideas around remasking/predictor-corrector sampling (Gat et al., 2024; Wang et al., 2025a) for inference-time scaling and/or planning (Ye et al., 2024; Huang et al., 2025; Zheng et al., 2023; Peng et al., 2025). EB-sampler (Ben-Hamu et al., 2025) focuses on efficiency, which is less studied for discrete diffusion models. Prior work like Park et al. (2024) also studied accelerating diffusion LLMs via a joint dependence perspective. There are also several recent works (Wu et al., 2025; Sahoo et al., 2025b) on enabling KV caches for discrete diffusion models, which will speed up both the inference and RL training processes. Most of these ideas are orthogonal to our contributions and can be easily integrated. We leave them as future work.

**RL for dLLMs**. Comparing to the extensive study on RLHF (Ouyang et al., 2022; Winata et al., 2025) and RLVR (Lambert et al., 2024; Zhang et al., 2025) for LLMs, there is noticably less study on dLLMs. Zekri & Boullé (2025) proposed to fine tune smaller scale discrete diffusion models with policy gradient methods, yet it requires full likelihood computation, which is hard and expensive for modern dLLMs. d1 (Zhao et al., 2025) proposes diffu-GRPO, which is the first work on training base dLLMs for reasoning with RL. Zhu et al. (2025) proposed a DPO-style (Rafailov et al., 2023) preference optimization method underpinning the development of LLaDA 1.5. Yang et al. (2025) proposed UniGRPO which utilizes random partial masking on completions instead of full masking. LLaDOU (Huang et al., 2025) trained a token-position aware sampler for enhanced dLLMs

reasoning performance. Concurrent to our work, wd1 (Tang et al., 2025) used weighted advantage function to improve upon $d1$ on the same math and planning tasks as ours, DiffuCoder (Gong et al., 2025) used coupled-GRPO to improve the base coding dLLMs, TraceRL (Wang et al., 2025c) also proposes to utilize online trajectory token latents, which is similar to our two times mean-field approximation, yet our parameterization based on time steps is more generalizable and flexible for different dLLM samplers.

## 6   DISCUSSION AND FUTURE WORKS

In this work, we propose DiFFPO – an RL pipeline for post-training dLLMs towards efficient reasoning. We showcase the effectiveness of our proposed algorithms in terms of better likelihood approximation, a unified off-policy RL perspective, and sampler joint training, which achieves better sample efficiency, superior performance, and enhancing the accuracy/efficiency tradeoff. We believe our contributions in the paper help push the boundary of RL for dLLMs post training research which is less explored.

There are several directions that can be pursued to improve or extend our work. It is interesting to understand the generalization behavior of the trained inference threshold; it is also possible to adapt our RL framework to a state (or group)-dependent inference threshold instead of being fixed for each prompt. In addition, we only test our model on LLaDA (Nie et al., 2025), it will be complimentary to see the performance on other open source dLLMs like Dream (Ye et al., 2025). We leave these investigations for future work.

## USAGE OF LLMS

We mainly utilized LLMs to help write code when conducting experiments to aid fast development of the training scripts of our training algorithms. We also use LLMs to generate some of the table and figure templates. To the best of our knowledge, all of these aforementioned do not violate the ICLR 2026 code of conduct.

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

# A MORE EXPERIMENTAL RESULTS

## A.1 BENEFITS OF TWO TIMES MEAN FIELD APPROXIMATION

We empirically measure the evolution of average KL divergence during the iterative decoding process of dLLMs across multiple reasoning benchmarks and the evidence that our proposed two times mean field approximation provides better approximation.

**Experimental Setup.** We use the LLaDA instruct version for inference and use confidence based remasking. We set the number of allowed unmasked token per time step to be 1. Temperature is set as 0 for deterministic generation. For each dataset, we choose 100 prompts to compute the average and set block size to 32. For simplicity, we only compute the results for the first block.

**Results on different datasets.** As showcased in Figure, for all datasets (GSM8K, Countdown, Sudoku and MATH), we have reduced avg KL divergence error by using two times mean field approximation.

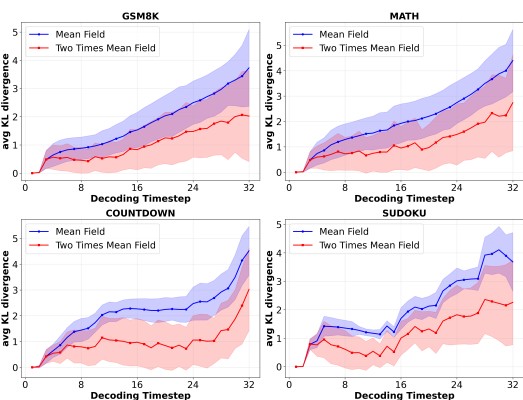

Figure 6: KL divergence between approximated likelihood and true likelihood.

## A.2 IMPORTANCE RATIO CLAMPING HYPERPARAMETER SWEEPING

**Importance ratio clamping.** We also study the effect of the role of importance ratio clamp hyperparameter $C$ and compute the average value after clamping for different clamping values. The results are showcased in Figure 7. Overall the smaller the clamp constant, the smaller the average of the clamped importance ratio. Despite that the constants in our clamp value search space is all able to control extreme ratio values, we still witness that the constants will have an non neglectable effect on the final obtained algorithm/model performance.

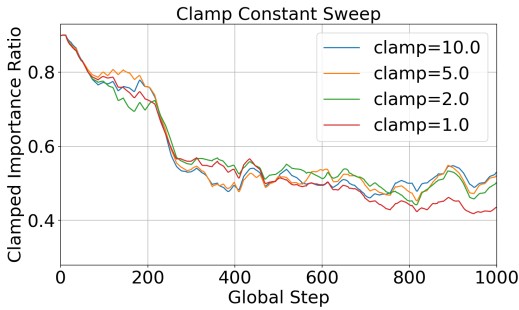

Figure 7: KL divergence between approximated likelihood and true likelihood.

