# OpenReview forum: "DiFFPO: Training Diffusion LLMs to Reason Fast and Furious via Reinforcement Learning"
_ICLR.cc/2026/Conference — Submitted to ICLR 2026_

### Official Review · Reviewer_fhVt · 2025-10-23

**Soundness:** 3
**Presentation:** 2
**Contribution:** 2
**Rating:** 4
**Confidence:** 2

**Summary:**

The authors propose DiFFPO, an RL framework for diffusion LLMs that replaces the intractable dLLM likelihood with a better surrogate via a two-times mean-field approximation and an importance-sampling correction. The method trains an efficient sampler by learning a prompt-aware entropy threshold γ(c), with optional joint training of model and sampler. The results show DiFFPO improving sample efficiency and accuracy on math and planning tasks while reducing NFEs versus d1 and fixed samplers.

**Strengths:**

1. The proposal of the surrogate-likelihood route for diffusion LLMs, the importance-sampling correction, and the prompt-aware sampler thresholds are all nice additions.

2. The paper figure and tables show  show consistent accuracy gains and better sample efficiency than d1.

3. The framing and equations are easy to follow.

**Weaknesses:**

1. The scope of the results (tested baselines + benchmarks) is quite small.
2. The method is not compared to other GRPO-like baselines (even though these are mentioned in the prior work section.
3. There are no experiments testing the method's stability & gains (the role of clip sweeps, ess/weight histograms, and a no-clip baseline.
4. Minor: The quality of the figures is low (e.g. fig 3).

**Questions:**

1. Can the authors explain the lack of benchmarks against comparative methods, and why comparing to d1 would suffice?

2. It would be worthwhile to see a light sweep of ε (and any hard cap C), report ESS/weight histograms per timestep, and include a no-clip (or self-normalized IS) reference, or have an intuition for what to expect.

3. Under what conditions do you expect DiFFPO to degrade? For example, 1. highly multimodal posteriors where the TTMF surrogate (single-τ conditioning) misses key latents, or 2. sparse/peaky rewards where clipping trims the few high-weight successes?

---

> ### Author Response · Authors · 2025-11-26
>
> Thank you for your detailed review and feedback for improving our manuscript. Below, we have addressed your questions as follows:
>
> ***Q1. Comparison to existing baselines***
>
> ***A1.*** Thank you for your suggestion on this. Among the papers we mentioned, most of them are concurrent work to us which has not been peer reviewed, such as DiffuCoder and wd1, and the rest of these works are working on different datasets to ours, making us hard to replicate and compare the performance. Yet we agree that comparison to these concurrent work could strengthen our findings, thus we compare with wd1 which focuses on the same set of experiments of ours, and our method still showcases evident better performance. Please find the comparison in our updated experimental sections, with also updated clear version of the original "Figure 3" - now Figure 4.
>
> ***Q2. Sweep of the caption constant***
>
> ***A2.*** Thank you for raising this question. We include the comparison of the learning curves and effective importance sampling ratio after being clamped by different caption constant in the Appendix A.2. In our experiments, we chose the best hyperparameters based on sweeping the constants and chose the best performant RL learning curve. We agree that this could strengthen the understanding of these hyperparameters.
>
> ***Q3. Conditions where DiFFPO will degenerate***
>
> ***A3.*** Thank you for your thoughtful question on this. Our primary contribution of our work for providing better RL training recipes for dLLMs from an off-policy RL perspective. We do believe that our method will not degenerate because of the aforementioned two conditions, as our method is pretty straightforward to apply multimodal cases by accessing intermediate latents of multimodal generations. For sparse rewards, we believe the tasks in our paper are already good examples, as the accuracy reward is only a 0-1 feedback signal, yet our algorithm still successfully enhances the reasoning capability of the models and show clear evidence in improving both efficiency and accuracy compared to the baseline approach.

---

### Official Review · Reviewer_hfeR · 2025-11-01

**Soundness:** 2
**Presentation:** 2
**Contribution:** 3
**Rating:** 4
**Confidence:** 4

**Summary:**

This work tackles the problem of RL fine-tuning for diffusion LLMs by proposing a modification of the GRPO objective used in prior work (namely d1 (Zhao et al., 2025)). The objective uses a more accurate token-level likelihood approximation (by evaluating the conditional likelihood $\pi_\theta(\hat{o}^t_i | c, z^{s}_i)$ at some intermediate partial completion $z^s_i$, for $t<s$ rather than just on the prompt as in d1), as well as an importance sampling correction to weight the GRPO advantage.  Additionally, a method is proposed to improve inference speed, whereby an input dependent threshold is learned  for use in Entropy Bounded sampling. The proposed policy gradient method is used to fine-tune both the model, and the sampling threshold predictor, for tasks such as Sudoku, Math, GSM8K and Countdown. An improvement in both task performance and inference efficiency (measured by NFEs) is demonstrated.

**Strengths:**

1. The experimental results reported are clear and convincing, both for the increase in reward, and the speed of inference (when tuning the Entropy Bounded threshold)
2. The analysis of the error in d1’s likelihood approximation is convincing (Figure 2 a, b)).
3. The idea of training the inference threshold in a sampler with RL is useful

**Weaknesses:**

1. Numerous spelling and grammatical mistakes throughout. The paper would benefit from more thorough proofreading.
    - Related to presentation Figure 2 c) is unclear, and includes symbols not defined
    - Figure 4 should indicate the threshold’s used when tracing out the frontier
2. For the efficiency comparison (eg. Figure 3), some plots should also be included where the x-axis is the number of calls to the denoiser, since the proposed two-times mean field approximation requires twice as many denoiser calls per gradient update (compared to d1’s mean-field approximation) when updating on a fixed set of trajectories. Without this, its difficult to judge whether the two times mean field approximation adds much benefit over the d1 baseline.
3. Several experimental details missing from the writeup - for instance there are no details provided for  batch sizes used, or whether remasking is performed (as described in the d1 paper) - these can be critical in evaluating whether the comparisons are fair
4. The impact of the importance weighting in eq 12 is unclear: we can sample trajectories from the surrogate (two-times mean field) policy and directly optimize it without any importance weights. How does this approach compare to using the importance weights?
    - Analogously, the importance weights can be directly applied to the d1 method. Comparing to this setup would help clarify the benefits of the two-times mean field approximation.
5. In terms of theory - Assumption 2 requires some sort of justification or argument (possibly empirical evidence). Theorem 3 is a straightforward corollary of Assumption 2, but it shifts the need for justification to Assumption 2.

For now I am recommending a weak reject, due to points 2 and 3 above.

**Questions:**

1. It would be useful to see a comparison against full likelihood evaluation to see the impact of the proposed corrections (eg. importance sampling), in a setting where it is feasible
2. Between table 2 and 3, why is the performance of the base model and d1 different?
3. Related to weakness 2 above:, one can extend the approach to an m-times mean field approximation. Is there some justification or evidence for why stopping at 2 is reasonable or adequate?

---

> ### Author Response · Authors · 2025-11-26
>
> We sincerely thank you for your detailed review and feedback. We also more than appreciate your kind words in terms of the contributions of our paper. Below, we have addressed your questions as follows:
>
> ***Q1. Improved presentation of Figure 2 and Figure 4***
>
> ***A1.*** We sincerely thank you for your suggestions on improving the writing of our manuscript. We provide more detailed descriptions in the manuscript about Figure 2 and also add the thresholds we choose in Figure 4 for obtaining the frontier. We have also fixed existing grammar mistakes as we can and have made our major changes in blue for your reference.
>
> ***Q2. Efficiency comparison with axis being number of neural network calls***
>
> ***A2.*** We sincerely thank you for your suggestion. We would like to clarify that in practice we stack the batch of prompt inputs and the batch of prompts with additional latents together to have one forward pass instead of calling the network twice. Since we only draw one additional latent, which only makes twice as the batch size, the GPUs can still easily accommodate the memory because of the larger batch size while improving the algorithm performance as showcased in our paper. In addition, for RL for training of existing large reasoning models, the most significant amount of time is spent on inference while the policy update usually takes minimum time (however if computing the full likelihood of dLLM, this will lead to significant amount of time). So this is the primary reason that our paper uses num samples as x-axis which we believe is a more convincing choice and indeed aligns with extra training time computation. Our algorithm also showcases clear advantage than the baseline method in reaching higher highest reward.
>
> ***Q3. Improved experimental details***
>
> ***A3.*** We sincerely thank you for your suggestion. We add more detailed experimental setups and explanations in the start of the experimental section, including the batch size and Top-k remasking mechanism. Our experiments have followed the setups in d1 for a fair comparison. We have marked the major part of our changes as below for the ease of reference.
>
> ***Q4. Ablation of the impact of importance sampling and two times mean field***
>
> ***A4.*** We sincerely thank you for your suggestion. We would like to refer to Table 1 where “+d1” means the d1 baseline, “+TwoTimesMean-Field” row indeed means applying two times mean-field to the d1 method. The final “+Importance Sampling” is indeed the d1 method with both two times mean-field and importance sampling, which showcases the clear benefits of both importance sampling and two times mean field. We add more detailed descriptions of the Table 1 and marked blue to have better presentations about what each row represents.
>
> ***Q5. Justification of Assumption 2***
>
> A5. We sincerely thank you for your suggestion. We add experiments to showcase the clearer comparison of resulting average KL divergence between our two times mean field approximation and original d1 likelihood approximation only conditioning on prompts. Please find Figure 2c and Appendix A.1. The results showcase the clear reduced error because of our two times mean-field approximation approach.
>
> ***Q6. Difference between Table 2 and Table 3***
>
> ***A6.*** Thank you for raising the question on this. The results and statistics in Table 1 are reported based on using Top k sampler as the sampler, and the results in Table 2 are based on EB sampler. As described in Table 3, we carefully chose the inference threshold of the EB sampler to the accuracy performance as good as even smaller values of thresholds in the RL curves.
>
> ***Q7. Extension to multiple times mean-field approximation***
>
> ***A7.*** We thank you for raising this insightful question. Actually we did run experiments with more than two times of approximation, e.g. three or four times, but we witnessed limited improvement beyond the two times baseline. Given that practically it will lead to more memory cost with more times, we only adopted the two times approximation which already showcases the clear advantage over the baseline d1 method.

---

### Official Review · Reviewer_XDTH · 2025-11-01

**Soundness:** 3
**Presentation:** 3
**Contribution:** 3
**Rating:** 6
**Confidence:** 3

**Summary:**

The paper presents a novel framework for fine-tuning diffusion LLMs, called DiFFPO. By utilising surrogate policies, a two stage likelihood approximation is adopted, which leads to improved sampling efficiency and task performance across a range of benchmarks.

**Strengths:**

The paper displays a high degree of novelty, and addresses the highly impactful and timely topic of diffusion LLMs.

The proposed method is theoretically well motivated, and this is clearly presented within the manuscript.

Experimental performance of the proposed model appears strong, showing substantial improvements in both speed and accuracy.

**Weaknesses:**

The main experimental results, while very promising and interesting,  are only presented with the best performing seed. This is not ideal as we then lack statistical uncertainty, and robustness. For example, the reader cannot be confident whether the improvement on MATH500 in Table 1 from 34.20 to 36.40 would persist if we had chosen a different set of random seeds.  RL can lead to a high variance across seeds and it is therefore especially important to fully present these findings.

Ideally it would have been informative to explore variations to the model architecture, or to  assess how this impact is expected to scale with the size of the model. Some comparisons to an 8b autoregressive LLM may also be informative.

**Questions:**

What metric was used to determine the best performing seed, is this on test performance or some validation metric?

What is the variance in empirical performance across runs with different seeds, and does this change significantly between the different methods?

---

> ### Author Response · Authors · 2025-11-26
>
> We would like to thank you for your detailed review and feedback.  We sincerely appreciate your kind words and finding our paper novel and interesting. Below, we have addressed your questions as follows:
>
> ***Q1. Performance across different random seeds***
>
> ***A1.*** Thank you for your suggestion on this. In terms of the randomness from generations of the obtained RL fine-tuned model, we add more detailed experimental setups and explanations at the start of the experimental section. In our experiments, we evaluate all obtained checkpoints using temperature 0. Thus for the same model, the generations are deterministic and they are the same across different seeds, leading to the same results on downstream evaluation metrics (accuracy).
>
> In terms of the randomness for RL training, We agree that performance across different random seeds could strengthen the significance of our algorithm, thus we have added the average reward performance with variance bar across different seeds of DiFFPO and its comparison with d1 baseline into the manuscript to showcase the clear advantage of our algorithms. We also added comparison to another baseline wd1.
>
> Our reported RL curve in the earlier manuscript is based on choosing the seed individually for each method that reaches the highest average reward in the learning curve. In the downstream outcome metrics, we indeed witness that there is a variance of performance across the best model for different seeds, yet we report the best performance for each method across seeds since generally only the best model is employed for usage.
>
> ***Q2. Comparison to the 8B autoregressive models***
>
> ***A2.*** Thank you for the suggestion on this. We have updated the statistics of several other representative AR models into the table as well for direct comparison.

---

### Official Review · Reviewer_YMFp · 2025-11-01

**Soundness:** 2
**Presentation:** 2
**Contribution:** 2
**Rating:** 2
**Confidence:** 3

**Summary:**

The paper studies the problem reinforcement learning with masked diffusion language models. The authors proposes DiFFPO, an off-policy RL approach that (i) optimizes a surrogate policy whose likelihood is tractable; (ii) improves that approximation with a two-times mean-field conditioning on an intermediate latent and adds importance-sampling (IS) correction; and (iii) trains the sampler itself by learning a prompt-aware entropy-bounded threshold and optionally jointly training policy + sampler. DiffPO is evaluated empirically by fine-tuning LLaDA-8B-Instruct on math and planning (GSM8K, MATH500, Sudoku, Countdown). The authors compare DiffPO with d1/diffu-GRPO baseline: with Top-k sampling, DiFFPO improves accuracy over base by +30.5 (Countdown) and +35.8 (Sudoku) points. With EB sampling and joint sampler training, DiffPO shifts the speed–accuracy frontier.

**Strengths:**

* The paper is generally well written. The problem is defined quite clearly and the proposed solution explained clearly.
* The 2MF approximation intuitively makes sense, and Theorem 3 shows it to be a tighter likelihood approximation than prompt-only (though I have some concerns about Assumption 2). The approximation is also handled in the objective with a IS correction.
* I like the idea of jointly training some sampler parameters instead of being fixed post-hoc to enable efficient sampling.

**Weaknesses:**

* I have a major concern about the empirical results. Specifically, the authors mention they select the best checkpoint from seeds for comparions. This can skew comparisons, putting the results in question. I suggest reporting the mean performance with some error estimates.
* Another concern is that DiffPO results are using top-k sampling and it seems that the baselines are not using it, this also makes the comparisons invalid.
* The paper misses some closely related prior work on RL fine-tuning of diffusion language models [1, 2, 3]. I believe a comparison to these baselines would be critical, as it is quite an active area.
* The experiments are also limited to a single base model. I think it would benefit the paper to have some experiments with another base model (e.g. https://github.com/DreamLM/Dream)

[1] Venkatraman et al., 2024. Amortizing intractable inference in diffusion models for vision, language, and control.

[2] Zekri and Boullé, 2025. Fine-Tuning Discrete Diffusion Models with Policy Gradient Methods.

[3] Tang et al. 2025. wd1: Weighted Policy Optimization for Reasoning in Diffusion Language Models.

**Questions:**

* Could you clarify the exact compute cost for the method and how it comapres to the baselines?
* Could you clarify the choice of the experimental setup, using the best seed and checkpoint?

---

> ### Author Response · Authors · 2025-11-26
>
> Thank you for your detailed review and feedback. We have updated our manuscript with more experimental results, explanations and comparisons, and we have marked our main changes by blue for ease of reading. Below, we have addressed your questions as follows:
>
> ***Q1. More experimental setups, and performance across different random seeds***
>
> ***A1.*** Thank you for your suggestion on this! We agree that reporting performances across different random seeds could strengthen the significance of our algorithm, thus we have added the average reward performance across different seeds of DiFFPO and its comparison with d1 baseline into the manuscript to showcase the clearer advantage of our algorithms.
>
> We also add more detailed experimental setups and explanations in the start of the experimental section. In our experiments, we obtain different checkpoints (because of different steps of RL training) for each random seed, and evaluate all the obtained checkpoints using temperature 0 (so the generation is deterministic) on downstream evaluation metrics (accuracy). Our reported RL curve in the earlier manuscript is based on choosing the seed individually for each method that reaches the highest average reward in the learning curve. Our reported downstream metric number is based on evaluating all those obtained checkpoints and choosing the best number, following the presentation in RL for LLM, see e.g. Deepseek R1.
>
> ***Q2. Choice of Sampler***
>
> ***A2.*** Thank you for pointing this out! We add more detailed explanations in the experimental setup since Top-k sampler is the default sampling method from official LLaDA repo and d1 also used Top-k sampler (although not explicitly mentioned in the paper). This can be justified by checking the official repo config in the [github link](https://github.com/dllm-reasoning/d1/blob/main/diffu-grpo/slurm_scripts/train.yaml), where remasking is chosen as “low_confidence”. We also add more details about our choice of hyperparameters, as our experiments follow the same setup as d1 for a rigorous comparison.
>
> ***Q3. Comparison to other baseline method***
>
> ***A3.*** Thank you for pointing out these missing references! We have added mentioned works in the related works session. Nevertheless, we would like to argue about the fundamental difference between our contribution and mentioned papers. For [1] it is designed for continuous space diffusion models, thus it is hard to be employed for dLLMs. For [2], we checked that the authors haven’t shared the codebase on the language tasks as in the latest version of its github, making us hard to replicate on large scale LLMs. In addition, it requires estimating the full likelihood ratio, which is computationally expensive for dLLMs and is one of the motivations of d1 and our methods. For [3], we agree that wd1 is also designed to improve d1. Primarily we didn’t compare to it because it’s a concurrent non peer-reviewed paper, and actually the improvement direction is orthogonal to our contribution as we are mainly improving the likelihood estimation, while wd1 used the same likelihood approximation as d1 and add weights on the advantages term. We do believe that the contributions of our work and wd1 can be complementary. Nevertheless, we provide= comparisons of our method with wd1, which showcases that DiFFPO outperforms both d1 and wd1 in the pure RL training (without training the sampler) setup.
>
> ***Q4. The exact computation cost for the method and comparison with the baseline.***
>
> ***A4.*** Thank you for your question on this! Theoretically we need an extra forward process of the neural network for computing the policy gradient since we need to compute the likelihood conditioning on our additional sampled latents. Yet in practice our extra computation cost is pretty minimal since we concatenate the prompt and prompt+additional latents into the same batch to enable one forward process of the model. Here that this clarifies our minimally added computational cost under the same setup of batch size.

---

### Meta-Review · Area_Chair_Zs6h · 2026-01-06

**Summary:**

The key decision factors were (i) fairness and robustness of the empirical evaluation, including seed selection, sampler parity, and compute accounting, (ii) whether the proposed two-stage likelihood approximation combined with importance sampling provides clear and consistent benefits over simpler baselines such as d1, and (iii) the breadth and stability of the baseline comparisons. Reviewers generally found the motivation and technical ideas interesting and timely, and several appreciated the analysis of likelihood approximation error and the idea of jointly learning sampler parameters. However, the overall assessment was strongly influenced by concerns about evaluation methodology and normalization. Although the rebuttal added multi-seed reward curves, clarified sampler choices, and expanded ablations, questions remained about whether the reported gains reflect robust improvements under fair, fully normalized comparisons.

**Reviewer Concerns:**

Addressed concerns: The authors clarified that Top-k/remasking settings are consistent with the d1 setup, added multi-seed averaged reward curves with variance to demonstrate training robustness, and expanded ablations disentangling the roles of the two-times mean-field approximation and importance sampling, including a comparison to wd1. The rebuttal also improved presentation clarity, justified key assumptions, and provided additional analysis supporting the likelihood approximation (e.g., KL/error diagnostics), as well as limited sweeps of clipping and sampler hyperparameters.

Remaining concerns: Despite these additions, some evaluation issues remain unresolved. Final downstream task accuracies are still reported using the best checkpoint or seed, which leaves residual fairness concerns, especially given the small absolute gains on some benchmarks. Compute normalization is incomplete, as denoiser calls or FLOPs for the two-times mean-field approximation are not fully matched or reported relative to d1, making efficiency comparisons difficult to interpret. Baseline coverage and stability diagnostics (e.g., broader baselines, more systematic sweeps, or variance analyses for final task metrics) are still limited, leaving uncertainty about the generality and robustness of the claimed improvements.

**Reviewer Scores:**

YMFp: ~4 XDTH: ~6 hfeR: ~4 fhVt: ~4.

---

### Decision · Program_Chairs · 2026-01-26

Reject